

**A GIS-based three-dimensional landslide generated waves height calculation**
**method**
Guo Yu[1], Mowen Xie [1,*], Lei Bu[1], Asim Farooq[2]
[1] School of Civil and Resource Engineering, University of Science & Technology
Beijing, Beijing 100083, China
[2] CECOS University, Peshawar Pakistan
[*] Corresponding author: Mowen Xie (mowenxie123@163.com)
**Abstract**: Combined with the spatial data processing capability of geographic
information systems (GIS), a three-dimensional (3D) landslide surge height calculation
method is proposed based on grid column units. First, the data related to the landslide
are rasterized to form grid columns, and a force analysis model of 3D landslides is
established. Combining the vertical strip method with Newton's laws of motion,
dynamic equilibrium equations are established to solve for the surge height. Moreover,
a 3D landslide surge height calculation expansion module is developed in the GIS
environment, and the results are compared with those of the two-dimensional Pan
Jiazheng method. Comparisons show that the maximum surge height obtained by the
proposed method is 24.6% larger than that based on the Pan Jiazheng method.
Compared with the traditional two-dimensional method, the 3D method proposed in
this paper better represents the actual spatial state of the landslide and is more suitable
for risk assessment.
**Key words**: landslide; waves height; grid column; GIS
**1. Introduction**
When a reservoir bank landslide body slides into the water, it will cause a waves
that can not only endanger the safety of passing ships and surrounding buildings but
also threaten the safety of the dam. Therefore, calculating the waves height is important
for evaluating the risks of landslides (Xu and Zhou, 2015).
The methods of calculating the landslide generated waves height can mainly be
divided into analytical method (Noda, 1970; Pan, 1980; Huang et al., 2012; Miao et al., 2011;
Di et al., 2008), numerical simulation method (Silvia and Marco, 2011; Montagna et al.,
2011), and physical modelling method (Ataie-Ashtiani and Nik-Khah, 2008; Cui and Zhu,
2011). Analytical method is widely used in engineering applications because of its


simple modelling processes, which has few requirements for engineers and high
precision.
The analytical method originated from Node (1970). Node proposed the waves
height calculation method on the basis of hydraulics. Since then, many scholars have
conducted more in-depth research. For example, Academician Pan Jiazheng of China
divided the landslide body into many two-dimensional (2D) vertical strips and
calculated the waves height by considering the horizontal and vertical movement of the
landslide. This method is called the Pan Jiazheng method (Pan, 1980). Huang et al. (2012)
improved the Pan Jiazheng method by considering the resistance of water and the
change in the friction coefficient. Miao et al. (2011) proposed a sliding block model
based on the 2D vertical strip method to predict the maximum waves height. The
American Civil Engineering Society recommends a prediction method of the waves
height (Di et al., 2008) that assumes the landslide results in the particle motion with a
centre of gravity, and Newton's law of motion is used to calculate the waves height.
The above methods are all 2D analysis methods. In the vertical strip method, the
calculation results will differ with the selection of the 2D section. The 2D analysis
methods cannot effectively simulate the actual spatial state of three-dimensional (3D)
landslide. Hu (Hu, 1995) proposed that the value obtained by 2D analysis method is
approximately 70% of the value based on 3D analysis method. To date, analytical
method based on the 3D landslide body model has not been studied by scholars.
Geographic information systems (GIS) is widely used in geotechnical engineering.
The most notable feature of GIS is that they can transform vector data into grid data
sets based on a grid column unit model (Xie et al., 2006a). Because of the high 3D spatial
data processing capability of GIS, many scholars have added geotechnical professional
models to their respective systems. For example, our research team established a 3D
limit equilibrium method based on GIS, and developed a slope stability analysis module
called 3Dslope (Xie et al., 2003a; 2003b; 2006b). Jia et al. (2015) proposed a slope stability
analysis method by coupling a rainfall infiltration model and 3D limit equilibrium
method within the GIS environment. Mergili (2014) combined GRASS GIS and the 3D
Hovland model to implement a 3D slope stability model capable of considering shallow
and deep-seated slope failures. Therefore, to develop a waves height calculation module
in GIS, it is necessary to first establish a force analysis model of the 3D landslide in
GIS.
Based on the spatial data processing capability of GIS, this paper applies the grid



column unit model to establish a 3D landslide model, and proposes a method for
calculating the waves height. Compared with 2D analysis methods, the 3D method
proposed in this paper better represents the actual spatial state of landslides.
Simultaneously, the resistance of the water is considered to improve the accuracy of the
calculation result. To make the calculation more convenient, an expansion module is
developed to calculate the waves height in GIS, and the feasibility of the module is
verified by a case study.
**2. GIS-based method of calculating the waves height**
**2.1. Grid column unit model**
For a slope, the representation of data is mainly in the form of vectors. These data
include but are not limited to slip surface, strata, groundwater, fault, slip, and other
types of data. These vector data layers can be converted to raster data layers using the
spatial analysis capabilities of GIS to form a grid data set. The grid data structure
consists of rectangular units. Each rectangular unit has a corresponding row and column
number and is assigned an attribute value that represents the grid unit (Xie et al., 2004).
Therefore, the slope can be divided into square columns based on the grid units to form
a grid column unit model, as shown in Fig. 1.

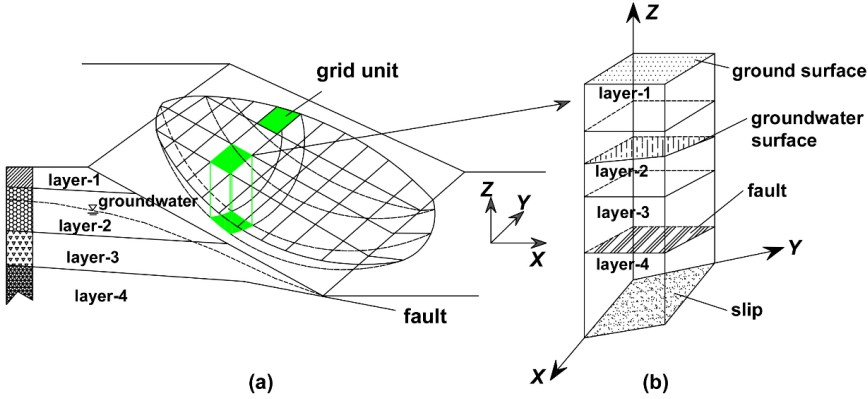


**Fig. 1.** Grid column unit model ((a) 3D view of landslide, (b) 3D view of one column).
**2.1. Force analysis**
First, we arbitrarily selected a grid column in a 3D landslide body, as shown in
Fig. 2. We can specify the forces acting on the grid column as follows.
(1) The weight of one grid column is $W$; the direction is the $Z$-axis; and the weight
acts at the centroid of the grid column.
(2) The resultant horizontal seismic force is $kW$, where $k$ is the "seismic




coefficient"; the direction of $kW$ is the sliding direction of the landslide; and the
resultant horizontal force acts at the centroid of the grid column.

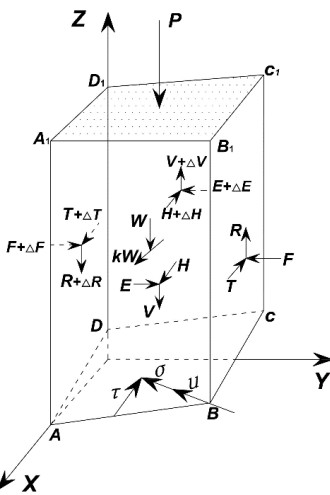


**Fig. 2.** Force analysis of one grid column.

(3) The external loads on the ground surface are represented by $P$; the direction of

$P$ is the $Z$-axis, and these external loads act at the centre of the top of the grid column.

(4) The normal and shear stresses on the slip surface are represented by $\sigma$ and $\tau$,

respectively. The normal stress is perpendicular to the slip surface, and the shear stress
is in the sliding direction of the landslide. The normal and shear stresses act at the
centroid of the bottom of the grid column.

(5) The pore water pressure on the slip surface is $u$.

(6) The horizontal tangential forces on the left and right sides of a grid column are

$T$ and $T+\triangle T$, respectively; the vertical tangential forces on the left and right sides of a
grid column are $R$ and $R+\triangle R$, respectively; the normal forces on the left and right sides
of a grid column are $F$ and $F+\triangle F$, respectively; the horizontal tangential forces on the
front and rear sides of a grid column are $E$ and $E+\Delta E$, respectively; the vertical
tangential forces on the front and rear sides of a grid column are $V$ and $V+\triangle V$,
respectively; and the normal forces on the front and rear sides of a grid column are $H$
and $H+\triangle H$, respectively. For convenience, the resultant force between columns in the
sliding direction of the landslide is defined as $\Delta D$.
**2.3. The spatial relationships among parameters**

Fig. 3 shows the 3D spatial relationships among parameters on the slip surface. $\theta$

is the dip of the grid column at the slip surface; $\alpha$ is the dip direction of the grid column
at the slip surface; $\beta$ is the sliding direction of the landslide; $\theta_r$ is the apparent dip of the
main inclination direction of the landslide; $\alpha_x$ is the apparent dip of the $X$-axis; and $\alpha_y$
is the apparent dip of the $Y$-axis.

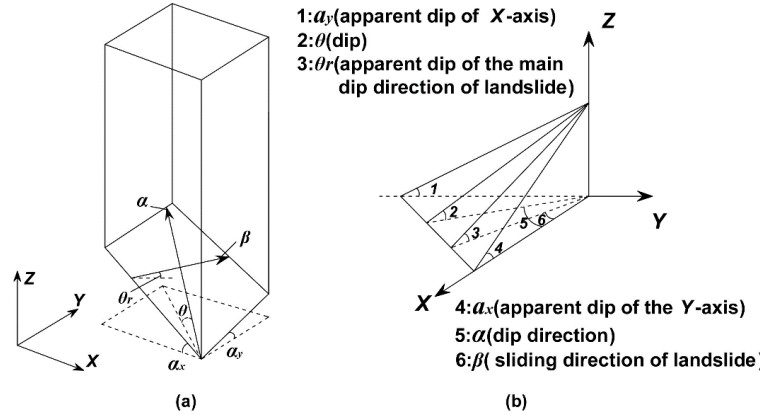


**Fig. 3.** 3D spatial relationships among parameters at the slip surface. ((a)
and (b) are the spatial relationships for 3D views of one grid column and
the coordinate system, respectively).
As shown in Fig. 3, the apparent dips of the $X$-axis and $Y$-axis are as follows.
$$\tan \alpha_x = \cos \alpha \tan \theta, \tan \alpha_y = \sin \alpha \tan \theta \qquad (1)$$
The slip surface area of one grid column is calculated by
$$A = cellsize^2 \left[ \frac{\sqrt{\left(1 - \sin^2 \alpha_x \sin^2 \alpha_y\right)}}{\cos \alpha_x \cos \alpha_y} \right] \qquad (2)$$
where *cellsize* represents the size of each grid column.
The apparent dip in the main inclination direction of the landslide is calculated as
follows.
$$\tan \theta_r = \tan \theta \left| \cos \left( \alpha - \beta \right) \right| \qquad (3)$$
The weight $W$ of the grid column is expressed as
$$W = cellsize^2 \sum_{m=1}^{n} h_m r_m \qquad (4)$$
where $m$ is the number of strata, $h_m$ is the height of each stratum, and $r_m$ is the unit
weight of each stratum. For the grid column units above the water, $r_m$ is calculated from
the natural unit weight. For grid column units under water, $r_m$ is calculated from the





buoyant unit weight.

The pore water pressure is obtained as follows (Zhang, 2016).

$$u = \frac{D}{\cos\theta}$$ (5)

where $D$ is the distance from the centre bottom of the grid column to the water surface.

When the sliding body enters the water, the resistance of the water is calculated as

follows (Chow, 1979).

$$G = \frac{1}{2} c_w \rho_f v^2 S$$ (6)

where $G$ is the resultant force of the resistance of the water to the sliding body; $c_w$ is the
viscous resistance coefficient, which is 0.18; $\rho_f$ is the buoyant density (g/m$^3$), taking the
average of all stratum; $v$ is the velocity of the landslide (m/s); and $S$ is the surface area
of the grid column in the water (m$^2$).
**2.4. Coordinate system conversion**

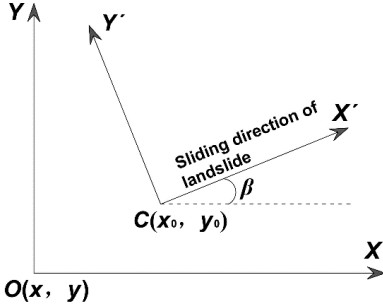


**Fig. 4.** Coordinate system conversion.

To facilitate subsequent calculations, the $XOY$ coordinate system was converted to

an $X'CY'$ coordinate system. The $X'$-axis direction was defined as the sliding direction
of the landslide. The right-hand rule determined the positive directions of the $Y'$- and
$Z$-axes. In addition, point $O$, i.e., the origin of the $XOY$ coordinate system, was
translated to point $C$ in the $X'CY'$ coordinate system, as shown in Fig. 4. The
transformation of the coordinates can be expressed as follows:

$$\begin{Bmatrix} x' \\ y' \end{Bmatrix} = \begin{bmatrix} \cos(90°-\beta) & \sin(90°-\beta) \\ -\sin(90°-\beta) & \cos(90°-\beta) \end{bmatrix} \begin{Bmatrix} x - x_0 \\ y - y_0 \end{Bmatrix}$$ (7)

where $x'$ and $y'$ are the coordinate values of the centre bottom of each grid column
in the $X'CY'$ coordinate system. $x$ and $y$ are the coordinate values of the centre
bottom of each grid column in the $XOY$ coordinate system; and $x_0$ and $y_0$ are the
coordinate values of point $C$ in the *XOY* coordinate system.
**2.5.  Dynamic equation based on grid column units**
We assume that all of the grid column units move continuously, do not separate in
the macroscopic dimension and remain vertical after sliding, as also assumed by Pan
Jiazheng (Pan, 1980). The force analysis of one grid column and the spatial relationships
among parameters at the slip surface are shown in Fig. 2 and Fig. 3, respectively.

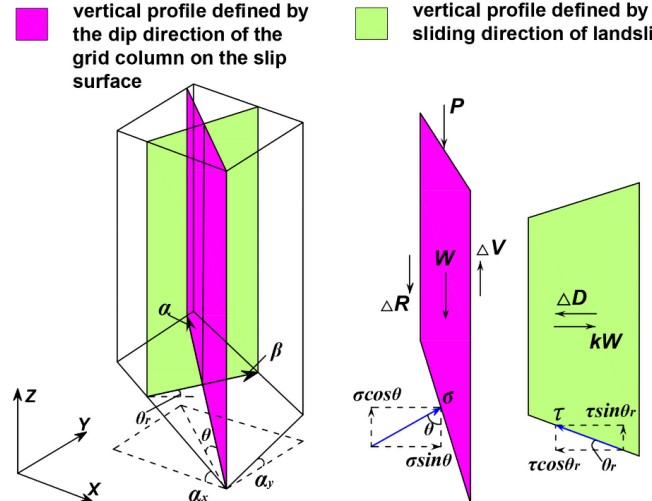


**Fig. 5.** Force analysis in the vertical direction and sliding direction of the landslide.
We arbitrarily selected a grid column unit (the grid column unit in row $i$ and
column $j$). According to Newton's laws of motion, dynamic equilibrium equations are
established in the sliding direction of the landslide and the vertical direction. The force
analyses in the sliding direction of the landslide and vertical direction are shown in
Fig. 5.
$$A_{i,j}\tau_{i,j}\cos\theta_{r_{i,j}} - A_{i,j}\sigma_{i,j}\sin\theta_{i,j}\cos\left(\alpha_{i,j}-\beta\right) - kW_{i,j} + \Delta D_{i,j} - G_{i,j} = \frac{W_{i,j}}{g}a_x \quad (8)$$
$$A_{i,j}\tau_{i,j}\sin\theta_{r_{i,j}} + A_{i,j}\sigma_{i,j}\cos\theta_{i,j} - W_{i,j} - P_{i,j} + \Delta V_{i,j} - \Delta R_{i,j} = \frac{W_{i,j}}{g}a_{y_{i,j}} \quad (9)$$
where
$$\tau_{i,j} = c_{i,j} + \left(\sigma_{i,j} - u_{i,j}\right)\tan\varphi_{i,j} \quad (10)$$
where $a_x$ and $a_{y_{i,j}}$ are the horizontal acceleration and vertical acceleration of the grid
column, respectively; $\varphi_{i,j}$ is the effective friction angle of the grid column at the slip



surface; $g$ is gravitational acceleration; $c_{i,j}$ is the effective cohesion of the grid column
at the slip surface; and $G_{i,j}$ is the resistance of water to the grid column. For grid column
units above water, $u_{i,j}$ is calculated by Eq. (5), and $W_{i,j}$ is calculated by taking the
natural unit weight. For grid column units under water, $u_{i,j}$ is 0, and $W_{i,j}$ is calculated
based on the buoyant unit weight.

According to this assumption, the horizontal acceleration $a_x$ of each grid column

unit is the same, and the vertical acceleration $a_{y_{i,j}}$ of each grid column unit varies. Pan
Jiazheng suggested that (Pan, 1980) there is a certain proportional relationship between
$a_x$ and $a_{y_{i,j}}$, that is, $a_{y_{i,j}}/a_x = \tan\delta_{i,j}$. $\delta_{i,j}$ is the horizontal inclination angle of the line
connecting the centre bottom of the grid column to the centre bottom of the next grid
column in the sliding direction of the landslide. The effect of vertical tangential forces
is ignored, namely, $\Delta V_{i,j} - \Delta R_{i,j} = 0$; therefore, Eq. (9) can be transformed as follows.
$$A_{i,j}\tau_{i,j}\sin\theta_{r_{i,j}} + A_{i,j}\sigma_{i,j}\cos\theta_{i,j} - W_{i,j} - P_{i,j} = \frac{W_{i,j}}{g}a_x\tan\delta_{i,j} \qquad (11)$$

The simultaneous Eqs. (10) and (11) can be obtained as follows.

$$\sigma_{i,j} = \frac{A_{i,j}\sin\theta_{r_{i,j}}\left(u_{i,j}\tan\varphi_{i,j} - c_{i,j}\right) + W_{i,j} + P_{i,j} + \dfrac{W_{i,j}}{g}a_x\tan\delta_{i,j}}{A_{i,j}\left(\sin\theta_{r_{i,j}}\tan\varphi_{i,j} + \cos\theta_{i,j}\right)} \qquad (12)$$

For the entire sliding body, the forces between the grid columns are internal forces,

that is, the resultant force is 0, yielding Eq. (13).
$$\sum_I\sum_J\Delta D_{i,j} = 0 \qquad (13)$$

By summing all the grid column units, the horizontal acceleration $a_x$ can be

determined by Eq. (8).
$$a_x = \left[\sum_I\sum_J\frac{A_{i,j}\tau_{i,j}\cos\theta_{r_{i,j}} - A_{i,j}\sigma_{i,j}\sin\theta_{i,j}\cos\left(\alpha_{i,j} - \beta\right) - kW_{i,j} - G_{i,j}}{W_{i,j}}\right]g \qquad (14)$$

Substituting Eqs. (10) and (12) into Eq. (14) yields the following equation.

$$a_x = \left[\sum_I\sum_J\frac{B_{i,j} + E_{i,j} - F_{i,j} - (G_{i,j}\tan\delta_{i,j}H_{i,j} - kW_{i,j})L_{i,j}}{W_{i,j}\tan\delta_{i,j}H_{i,j}Q_{i,j}}\right]g \qquad (15)$$
where
$$B_{i,j} = A_{i,j}\cos\theta_{r_{i,j}}\left(u_{i,j}c\tan\varphi_{i,j} - c_{i,j}^{\ 2}\right) \qquad (16)$$


$$E_{i,j} = A_{i,j} \cos\left(\alpha_{i,j} - \beta\right) \sin\theta_{r_{i,j}} \sin\theta_{i,j} \left(u_{i,j} \tan\varphi_{i,j} - c_{i,j}\right) \qquad (17)$$
$$F_{i,j} = \left[\cos\theta_{r_{i,j}} \tan\varphi_{i,j} - \sin\theta_{i,j} \cos\left(\alpha_{i,j} - \beta\right)\right]\left(W_{i,j} + P_{i,j}\right) \qquad (18)$$
$$L_{i,j} = c_{i,j} + \sin\theta_{r_{i,j}} \tan\varphi_{i,j} \qquad (19)$$
$$H_{i,j} = \cos\theta_{r_{i,j}} \tan\varphi_{i,j} - \sin\theta_{i,j} \cos\left(\alpha_{i,j} - \beta\right) \qquad (20)$$
$$Q_{i,j} = c_{i,j} + \sin\theta_{r_{i,j}} \tan\varphi_{i,j} \qquad (21)$$
**2.6. Calculation of the sliding velocity**

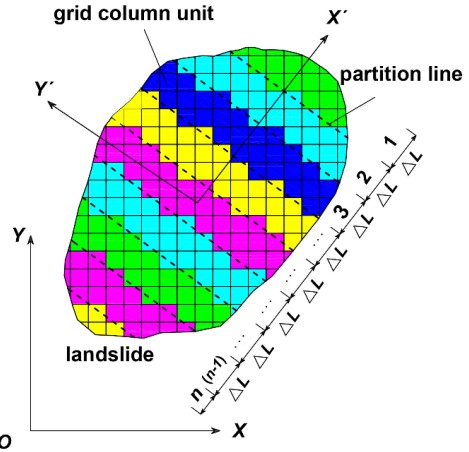

**Fig. 6.** Rasterization and partitioning of landslides.
The steps in calculating the landslide sliding velocity are as follows.
(1) Using the spatial analysis capability of GIS, the landslide body is rasterized,
and the size of the grid column unit $(i, j)$ can be set to an arbitrary square. A partitioning
line is drawn from the bottom to the top of the landslide every $\Delta L$ in the sliding direction
of the landslide, and the resulting regions are numbered zone 1, zone 2, zone 3, ..., zone
(n-1), zone n. Each partition includes a number of grid column units, and the length of
zone n is less than or equal to $\Delta L$, as shown in Fig. 6. For a grid column unit that is not
completely contained within a partition, if the area within the partition is greater than
half of the total area, the unit is divided into that partition; otherwise, the unit is divided
into the next partition.
(2) For each grid column unit, the parameters required in Eq. (15) are calculated.
(3) $t_0$ is the starting point of when the landslide body begins to slide, and $t_0=0$.
When the landslide body moves distance $\Delta L$ sequentially in the sliding direction of the





landslide, the corresponding time is recorded as $t_1$, $t_2$, $t_3\ldots t_n$, and the corresponding
velocity is expressed as $v_{x1}$, $v_{x2}$, $v_{x3}\ldots v_{xn}$.

(4)The horizontal acceleration at $t_0$ can be calculated by Eq. (15) and is denoted

as $a_{x0}$, and the velocity at time $t_0$ is zero. After sliding distance $\Delta L$ is reached, the
following equations can be obtained.

$$v_{x1} = \sqrt{2a_{x0}\Delta L} \tag{22}$$

$$t_1 = t_0 + \sqrt{\frac{2\Delta L}{a_{x0}}} \tag{23}$$

(5) At $t=t_1$, the landslide body has horizontally moved by a distance $\Delta L$ in the

sliding direction of the landslide, zone 1 has slipped form the sliding surface. The
horizontal acceleration $a_{x1}$ at $t_1$ is still calculated by Eq. (15). Unlike $t_0$, the weight for
zone (n-1) changes to the weight for zone n, and the weight for zone (n-2) becomes the
weight for zone (n-1), and so on (at this time, there is no grid column for zone n). After
$a_{x1}$ is calculated, the following can be established.

$$v_{x2} = \sqrt{2a_{x1}\Delta L + v_{x1}^{2}} \tag{24}$$

$$t_2 = t_1 + \frac{v_{x2} - v_{x1}}{a_{x1}} \tag{25}$$

(6) The calculation is continued in turn. When the obtained horizontal acceleration

is negative, the maximum velocity can be obtained. Finally, $a_x$ and $v_x$ in the calculation
process can be plotted as respective curves versus the sliding time.
**2.7 Calculation of the waves height**

The China Institute of Water Resources and Hydropower Research proposed an

empirical formula for waves height calculation (Zhong et al., 2007). In the formula, the
main factors that affect the waves height are the sliding velocity and volume of the
landslide. The formula for calculating the maximum waves height is as follows.

$$\xi_{\max} = d\,\frac{v_m^{1.85}}{2g}V^{0.5} \tag{26}$$

where $\xi_{\max}$ is the maximum waves height (m); $d$ is the comprehensive influence
coefficient, with an average value of 0.12; $v_m$ is the maximum sliding velocity (m/s); $V$
is the volume of the landslide body in the water ($m^3$); and $g$ is gravitational acceleration,
which equals 9.8 m/s$^2$.



The formula for calculating the waves height at different distances from the

landslide body is as follows.
$$\xi = d_1 \frac{v_m^{\ n}}{2g} V^{0.5} \tag{27}$$

where $\xi$ is the waves height at a distance of $L$ metres from the landslide body (m); $n$ is
the calculation coefficient, which is 1.4; and $d_1$ is the influence coefficient related to
distance $L$, which is determined by the following formula.
$$d_1 = \begin{cases} 0.5 & , \ (L \le 35) \\ 6.1274L^{-0.5945}, & (L > 35) \end{cases} \tag{28}$$

**3. Program implementation**

Combined with the waves height calculation method, an expansion module was

developed based on component object model (COM) technology in the ArcGIS
environment. Fig. 7 illustrates the computational process.

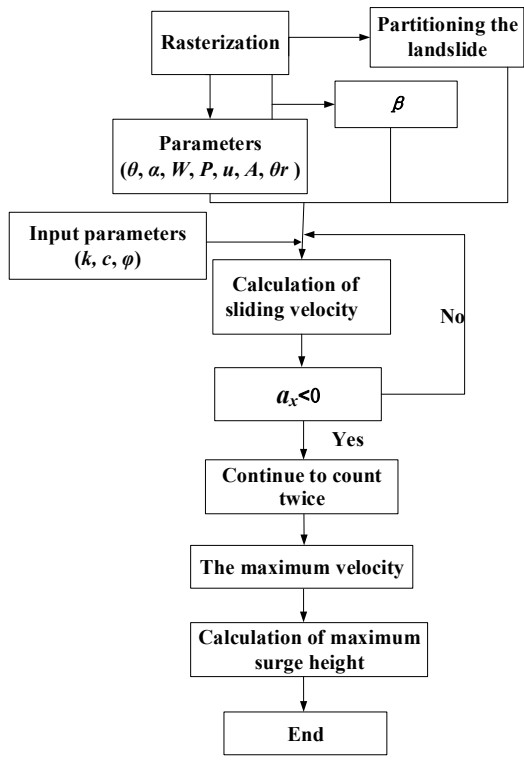

**Fig. 7.** The computational process.

**4. Case study**



### 4.1. Overview of the project

The Kaiding landslide is approximately 14.5 km away from the dam of the Houziyan hydropower station in Sichuan, China. The length of the landslide along the river is approximately 490 m, the top elevation is 2080 m, the bottom elevation is 1754 m, and the volume is approximately 4.5 million cubic metres. Plan and section views are shown in Fig. 8 and Fig. 9, respectively.

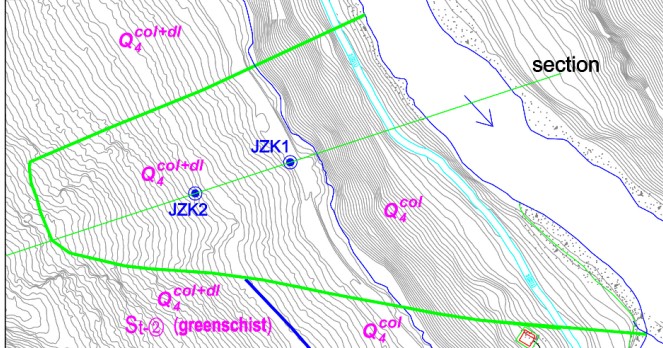

**Fig.8.** The plan view of the Kaiding landslide.

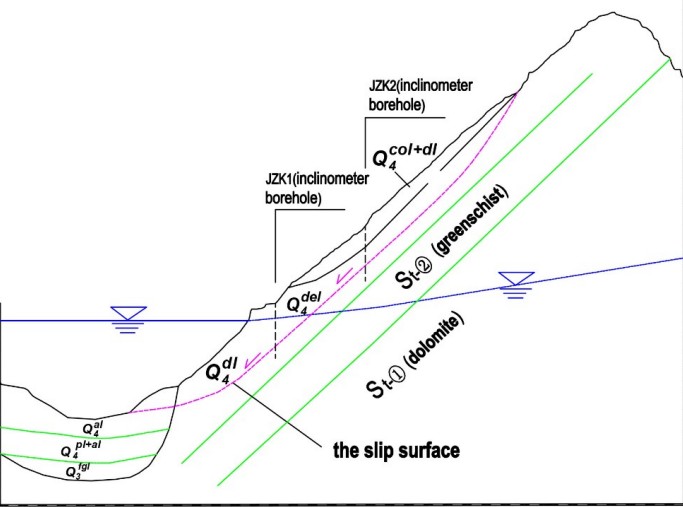

**Fig. 9.** The section view of the Kaiding landslide.

### 4.2. Calculation of the sliding velocity

The unit size of a grid column is 5 m×5 m, and $\Delta L = 10$ m. The internal friction angle $\varphi$ at the slip surface is 22.8°, the natural unit weight is 18.84 kN/m$^3$, the buoyant unit weight is 19.43 kN/m$^3$, the buoyant density is 2.11×10$^6$ g/m$^3$, and the elevation of the reservoir water level is 1810.3 m. When the landslide body slides, the effective cohesion $c$ at the slip surface will decrease to 0, that is, $c=0$ (Pan, 1980). Using this

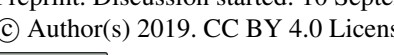


method and Pan Jiazheng's 2D method, the acceleration and velocity curves with the
sliding time can be obtained, as shown in Fig. 10 and Fig. 11, respectively. The
calculation results are shown in Table 1.

**Table 1** Calculation results

| The Pan Jiazheng method | | | The proposed method | | |
|---|---|---|---|---|---|
| $t$(s) | $a_x$(m/s²) | $v_x$(m/s) | $t$(s) | $a_x$(m/s²) | $v_x$(m/s) |
| 0 | 0.84 | 0 | 0 | 1.25 | 0 |
| 3.65 | 1.52 | 5.48 | 3.30 | 1.84 | 6.07 |
| 5.21 | 1.21 | 7.35 | 4.70 | 1.50 | 8.17 |
| 6.47 | 0.94 | 8.49 | 5.83 | 1.19 | 9.52 |
| 7.61 | 0.66 | 9.17 | 6.84 | 0.88 | 10.40 |
| 8.68 | 0.34 | 9.49 | 7.78 | 0.60 | 10.96 |
| 9.73 | 0.02 | 9.51 | 8.67 | 0.28 | 11.21 |
| 10.81 | -0.35 | 9.13 | 9.57 | -0.08 | 11.14 |
| 11.95 | -0.71 | 8.33 | 10.48 | -0.45 | 10.73 |
| 13.28 | -1.27 | 6.74 | 11.45 | -0.90 | 9.86 |

The calculation results indicate that the maximum velocity obtained by the
proposed method is 11.21 m/s, the starting acceleration is 1.25 m/s², and the sliding
time required to reach the maximum velocity is 8.67 s. In comparison, the maximum
velocity obtained by the Pan Jiazheng method is 9.51 m/s, the starting acceleration is
0.84 m/s², and the sliding time required to reach the maximum velocity is 9.73 s.
Comparing the results of the proposed method with those of the Pan Jiazheng
method, the maximum velocity of the proposed method is 15.2% higher than that
calculated by the Pan Jiazheng method, the starting acceleration is 32.8% higher, and
the sliding time required to reach the maximum velocity is 1.06 s short.

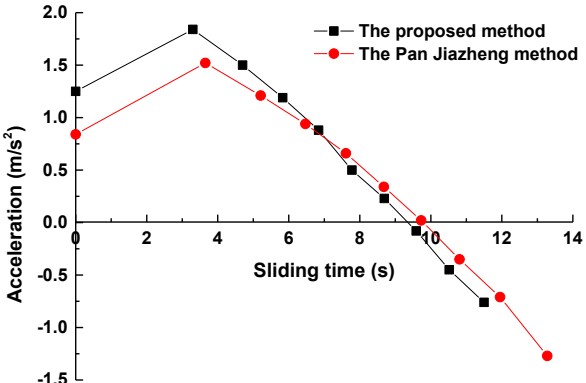

**Fig. 10.** Horizontal acceleration curve with the sliding time.



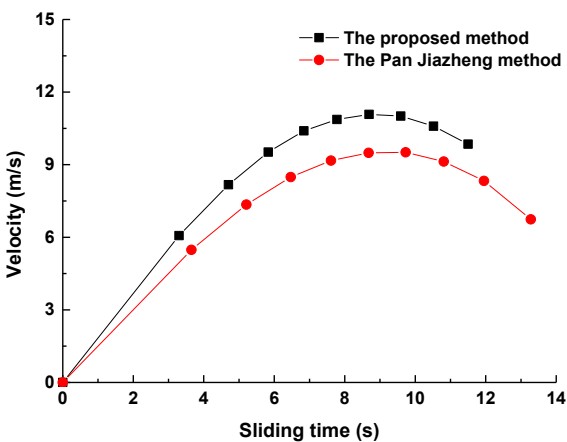

**Fig. 11.** Sliding velocity curve with the sliding time.

**4.3. Waves analysis**
According to the most dangerous working conditions, it is assumed that the
landslide body all slips into the water. The volume $V$ of the landslide body under water
is $340 \times 10^4$ m$^3$. According to Eqs. (26) and (27), the maximum waves height obtained
by the proposed method is 9.66 m, and the waves height at the dam site is 0.56 m. The
maximum waves height obtained by the Pan Jiazheng method is 7.28 m and the waves
height at the dam site is 0.44 m.
The landslide is approximately 14.5 km from the dam, the crest elevation is
1847.02 m, and the elevation of the reservoir water level is maintained at 1810.3 m.
When the waves height at the dam site is 0.56 m, water will not flow over the dam crest
and the safe operation of the dam will not be affected.
The maximum waves height obtained by the proposed method is 24.6% larger than
that based on the Pan Jiazheng method, and the waves height at the dam site obtained
by the proposed method is 21.4% larger than that based on the Pan Jiazheng method.
The calculations indicate that the results of the 2D method are smaller than those
of the 3D method. Compared that of the 2D method, the computational model of the
3D method better represents the actual spatial state of the landslide. As an analytical
method, the 3D model in this paper is more suitable than the 2D model.
**5. Conclusions**
Combined with the powerful spatial analysis ability of GIS, a 3D landslide force
analysis model based on grid column units was established. The dynamic equilibrium
equation for calculating the sliding velocity of a 3D landslide was derived to calculate




the waves height by combining Newton's laws of motion. To make the calculation more
convenient, an expansion module is developed to calculate the waves height in GIS,
and the feasibility of the module is verified by a case study.
Through calculations based on the case study, the maximum waves height
calculated by the 3D method proposed in this paper is 24.6% larger than that based on
the 2D Pan Jiazheng method, and the sliding time required to reach the maximum
velocity is shorter by 1.06 s. The calculations indicate that the results of the 2D method
are smaller than those of the 3D method.
Because the Pan Jiazheng method is based on a 2D section, the calculation results
will vary with the selected section. In this paper, the 3D landslide body model based on
grid column units is used to overcome the above shortcomings, and the calculation
model better represents the actual spatial state of the landslide body. Therefore, the
proposed method is more suitable for practical risk assessment.

**Data availability:** All data generated or analysed during this study are included in this
published article.

**Author contributions:** G.Y. and M.X. conceived of the presented idea. G.Y.
implemented the algorithm, and developed the theory. G.Y., M.X., and A.F. revised the
manuscript critically. G.Y. and L.B. finished the programming work. A.F. checked the
language.

**Competing interests:** The authors declare that they have no conflict of interest.

**Acknowledgment:** I would like to express my sincere gratitude to Prof. Mowen Xie,
Lei Bu, and Asim Farooq for their motivation and for providing me access to their
immense knowledge during this research work. This work was supported by the
National Natural Science Foundation of China [grant numbers 41372370].






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
