# Peer review of "A GIS-based three-dimensional landslide generated waves height calculation"

_Natural Hazards and Earth System Sciences, 2019_

## Referee Comment (RC1) · Anonymous Referee #1 · 26 Sep 2019

General comments

In the present paper a new method is presented to calculate the height of waves generated by 3D landslides. This method has been implemented and integrated in a geographic information system (ArcGIS).

This is an interesting topic which fits with the scope of the journal. Moreover, the paper is generally well organized, clearly written and easy to read.

However, the article is a bit superficial in many aspects, and a major revision is needed before it can be considered for publication.

Specific comments

[Figure]

Line 36. The waves height calculation method by Noda (1970) should be better presented and explained.

Lines 80-82. In my opinion, this sentence is obvious for scientists and engineers, who work with GIS.

Figure 2. The figure is not clear. Too many small arrows, too many symbols. Please, enlarge the figure or the arrows, to make single contributions more understandable.

Line 97. What are "the external loads"? This must be better explained.

Lines 98, 102 and 139. "centre" and "centroid" are the same? If so, please unify.

Line 103. Please, specify that u is directed as sigma.

Lines 104-112. It is not clear what the authors mean with "left and right sides". It would be better to refer to the coordinate system of Fig.2. In this way "left and right sides" becomes "vertical face at y = 0" and "vertical face at y = Dy", being Dy the size of the grid column along Y. Moreover, usually the stress is associated to the side at y = 0 and its increment to the side y = Dy (for example T at the vertical face at y = 0 and T+DT at the vertical face at y = Dy): in Fig.2 it is the opposite.

Line 138. u is declared as a pressure, so it should be a force per unit area and not a distance as D is. Moreover, symbol D is very similar to symbol DD and it should be changed to avoid misunderstanding.

Line 144. In the literature there is some variability in the choice of the value for the viscous resistance coefficient. The choice of the authors must be better motivated.

Paragraph 2.4. This paragraph is obvious and banal and therefore it can be removed. Suffice it to keep Fig.4 and to say that the analysis of each column was done with respect to a coordinate system X'CY', being X' the sliding direction of the landslide.

Lines 180-183. These two phrases are basically a repetition of concepts already mentioned before.

Paragraph 2.7. The wave height is not evaluated in a new and original way, but it descends from the application of a literature formula (Zhong et al., 2007). The title of the paper is misleading, because it makes you think of a new technique to evaluate waves height. Instead, as far as I understand, the authors present a new technique to evaluate landslide velocity and they apply a literature formula to calculate waves height. This must be better clarified in the title and in the Introduction. Moreover, the formula by Zhong et al. (2007) should be better presented and discussed, as it becomes part of the final GIS.

Paragraph 4. In the case study, the presented method has been applied together with Pan Jiazheng method.

To compare them, Pan Jiazheng method should be better presented and explained. What are the main differences between the two approaches, besides the fact that one is 2D and the other one is 3D?

Moreover, the results are presented, but it is not clear to the reader why this new method is better than the previous one. What are the benefits?

Which terms are here considered, which have been previously neglected?

And how can this new approach still be improved?

Almost 25% of the cited references are in Chinese. NHESS is an international journal, with readers from all over the world. In my opinion, it would be preferable to refer to papers written in English, as far as this is possible. Please, try to find some alternative references in English, even by the same authors.

Moreover, the reference list should be extended.

Technical corrections

Please, check all the references:

Line 28. Is this "Xu and Zhou, 2015" as it appears in the text or perhaps "Xu et al.,

2015" as it appears in the References?

Lines 31 and 46. Is this "Di et al., 2008" as it appears in the text or perhaps "Di and Sammarco, 2008" as it appears in the References?

Line 31 and 380. "Silvia and Marco" sound as given names, please check the correct reference.

Line 36. Please, change in "Noda".

Line 51. Please, change in "Hu (1995)".

Line 62. Is this "Mergili (2014)" as it appears in the text or perhaps "Mergili et al. (2014)" as it appears in the References?

Please, check the character size of the references, which seems to be smaller than the rest of the text (in particular lines 30-33).

Please, check the alphabetical order: Mergili et al. (2014) should come before Miao et al. (2011); Zhang (2016) should come before Zhong et al. (2007).

Lines 346-349. Please, change the text to impersonal form.

---

## Referee Comment (RC2) · Anonymous Referee #2 · 29 Oct 2019

This manuscript provides the calculation method for 3D landslide surge height using spatial data processing capability of GIS. The reviewer believes that the subject of this manuscript is interesting but the following points should be clarified to recommend for publication.

1. The reviewer found some English syntax problems in the manuscript. The manuscript should be edited by English speaking natives.

2. In this manuscript, the authors compared the proposed approach to Pan Jiazheng method. In order to understand the improvement of the proposed approach, the brief explanation about the basic equations and calculation procedures of Pan Jiazheng should be introduced in Section 2.

3. Section 3 is too short. The explanation about Fig. 7 may be included or this part should be included in section 2.8.

4. As the authors mentioned, the Pan Jiazheng method had been improved by Huang et al (2012) and Miao et al (2011). But the authors compared the results of the proposed approach only with the results of Pan Jiazheng, but not for Huang et al (2012) and Miao et al (2011). The reviewer believes that the results of the proposed approach also should be compared with the improved approached to show this approach is better than the previous ones.

5. In order to find out the feasibility of the proposed approach, the analysis results of the proposed method should be compared actual data of the landslide surge. However, the authors only compared the results with the those from Pan Jiazheng, which are also the calculation results. Since the authors mentioned in line 315-316 of the manuscript 'the computational model of the 3D method better represents the actual spatial state of the landslide', the authors should provide the comparison of the analysis results between the proposed approach and actual data.

6. The reviewer believes that one of the important part of the research paper is the discussion about the limitation of the proposed approach and the possibility of further improvement. So the reviewer recommends to include the discussion part in the manuscript.

---

## Referee Comment (RC3) · Anonymous Referee #3 · 2 Nov 2019

Dear authors, Dear editor,

I have reviewed the aforementioned manuscript (nhess-2019-230). The study presents a GIS-based approach to calculate waves from 3D-approximated landslides. The study is very interesting, partly well written and the research objective is relevant. However, results and discussion are not or very rudimentary presented. I suggest the authors to perform a **major revision** of their manuscript. Please find general and specific comments below.

**General comments** - The study fits in the scope of the journal. - Although the title comprises the topic of the study, it reads staccato-like caused by the many nouns listed one after the other. I believe rephrasing the title a bit so that it appears to be more fluent to the reader would be beneficial for recognising the study. - The manuscript is well

[Figure]

structured but massively lacks of detailed presentation of results and a comprehensive discussion of the findings.

**Specific comments** - Figure 2 looks a bit messy and should be improved. - In my perception, Section 2.1 should be moved to an appendix. - There is a Section 2.1 and 2.3 but no 2.2. - Line 164: Just mention the reference and the year, not the full name. - Line 186: See previous comment. - Line 191-208: This should be moved in an appendix. - Line 209: A section should not start with a figure. - Line 214-216: How is this performed in ArcGIS? - Section 4.1 should be extended and renamed in my opinion. - Based on the results (?) presented and the entire manuscript, the reader is not able to comprehend the conclusion made by the authors.